# Large Language Models for Low-Resource Languages: A Plan for Te Reo Māori

Luca Blaauw Fossen[*1]

[1] lf226@students.waikato.ac.nz

## 001 Abstract

Large Language Models perform remarkably well on high-resource languages, but lag behind for low-resource and Indigenous languages [1]. This has prompted several language communities to create specialized fine-tuned models for their language. This extended abstract presents an early-stage plan to develop the first sovereign Māori large language model. This plan includes curating high-quality Māori text datasets, constructing culturally relevant benchmarks, and performing continual pre-training and instruction-tuning of open-weight foundation models. This work will be done under Māori expert oversight and community participation from Māori language speakers and iwi (tribes), as well as the CARE principles of Collective Benefit, Authority to Control, Responsibility, and Ethics [2]. At this stage, corpus creation, model choice, and evaluation methods remain under exploration.

## 1 Motivation & Significance

Large language models have recently become remarkably popular in AI research and society in general, due to their impressive performance across a wide range of NLP tasks. Today, LLMs are usually trained as general-purpose assistants, making them easily accessible to laypeople. However, due to differing amounts of available training data, they exhibit a significant performance difference between high-resource languages (such as English, Chinese, Spanish, etc.) and low-resource languages (including Basque, Te Reo Māori, Sámi, Xhosa, etc.). Preliminary evidence from multilingual translation benchmarks suggests that performance on Māori lags behind English by 10-15 chrF points[3], and some languages lag behind by up to 40 chrF points [4].

This performance gap raises a pertinent concern: As large language models continue to advance, speakers of indigenous and smaller languages risk being left with AI and NLP systems that are much less capable than their high-resource counterparts. Reducing such performance gaps is essential to avoid side-lining indigenous languages in society.

---

*Supervision from Albert Bifet, Te Taka Keegan, Johan Barthelemy and Samia Touileb.

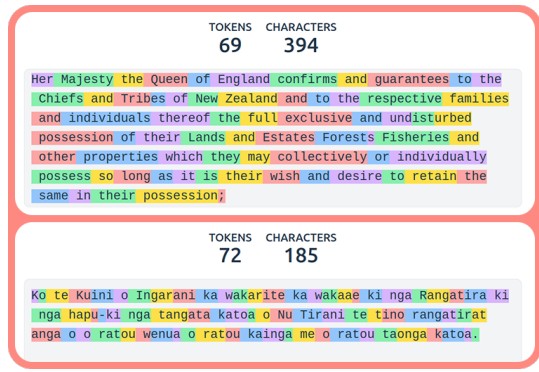

**Figure 1.** The English and Māori versions of the treaty of Waitangi (Article two, first clause), highlighted by GPT-4 tokenizer. The Māori version shows over 2X higher token density, consuming more tokens despite having less than half the character count of the English version.

## 2 Related Work

Various local initiatives around the world have emerged to create fine-tuned models tailored to their own languages.

In the Basque country, fine-tuning Llama for the Basque language has produced impressive models competitive with closed-source frontier models across Basque tasks [5], as well as a Basque-adapted evaluation suite called BasqueGLUE [6].

In Singapore, attempts to create models for South East Asian languages have outcompeted GPT-3.5 on South East Asian language tasks [7]. In Norway, NORA.LLM has trained models for Norwegian and Sámi, performing well against similarly sized models [8].

Existing Māori deep learning NLP work focuses on speech, likely due to the prevalence of Māori radio broadcasts and thus audio data. For example, Te Hiku Media have developed ASR technology as well as a speech benchmark for Māori that reflects the language's unique characteristics and diversity of speakers. [9].

To our knowledge, no autoregressive text modeling efforts for Māori have been published, and apart from massively multilingual datasets like FLO-RES200 [10], there exist no publications on benchmarks designed for Māori LLM evaluation.

Some scholars warn that machine learning may

harm Indigenous languages if applied without community control or linguistic rigor, and can also lead to data poisoning, harming future data collection efforts [11]. We welcome these concerns, and commit to remaining aligned with CARE principles by ensuring Māori authority over data use, community participation in corpus creation and evaluation, and collective benefit through open, culturally governed model development. We also recognize the danger of data poisoning as a real and substantial challenge to low-resource languages, and because of this, we will devote substantial efforts towards developing methods for assessing the quality and authenticity of indigenous text, as well as developing watermarking technologies for our own models.

# 3 Proposed Work

In order to create an LLM for Māori, we need a high-quality corpus, instruction tuning dataset, and culturally relevant benchmarks. These resources do not yet exist, so an initial step for us is to create them.

## 3.1 Data Curation

Based on initial corpus analysis, we have identified over 500 million Māori words across public and private datasets. These include news, [12], parliamentary transcripts [13], web corpora like HPLT [14]. However, preliminary filtering suggests that a substantial portion of data may warrant exclusion due to low quality or corpus duplication, so the final usable corpus size is still being determined. Additional private data collections under Māori custodianship such as transcriptions of radio broadcasts, books, and private data collections are also yet to be collected and counted.

We will create instruction fine-tuning datasets directly from Māori speakers through participatory approaches, as well as by adapting existing datasets.

## 3.2 Data and compute efficiency

Many higher-resource languages have hundreds of billions or even trillions of available tokens. Because Māori resources are comparatively limited, methods of efficient data and compute utilization will be investigated. On the data side, we will explore generating synthetic text from Māori knowledge graphs, as well as common techniques like back-translation and paraphrastic augmentation. On the compute side, we will explore the suitability of second-order optimizers (e.g. Sophia [15]) for language modelling, as well as standard approaches like parameter-efficient fine-tuning (PEFT) and model quantization.

## 3.3 Tokenizer adaptation

Preliminary experiments indicate that Māori text is tokenized up to twice as inefficiently as English by current models (Figure 1). This presents an opportunity to analyze and adapt tokenizers to reduce token fertility and improve subword coverage for Māori orthography.

## 3.4 Model evaluation

Model evaluation will rely on both automatic metrics (perplexity, BLEU/chrF++ on custom benchmarks) as well as human assessments of fluency and cultural integrity.

For evaluation, we plan to create MāoriGLUE, a General Language Understanding Evaluation suite created to serve the Māori language community, inspired by the original GLUE [16] and BasqueGLUE [6]. We will consult Te Hiku Media and collaborate with the Māori Language Commission [17] to create this evaluation suite.

We plan to involve the Māori language community through participatory approaches to collect training examples, build benchmarks and perform model evaluation. The entirety of this work will be carried out with guidance from Māori representatives, and will be done in accordance with CARE principles.

# 4 Expected Contributions

We expect to produce the first sovereign language model for Te Reo Māori. In creating this, we will also produce datasets and other artifacts to further the development of NLP for Māori, as well as results that can inform and guide the development of LLMs and NLP technology for other low-resource languages.

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
