# OpenReview forum: "Large Language Models for Low-Resource Languages: A Plan for Te Reo Māori"
_NLDL.org/2026/Abstracts_Track — NLDL 2026 Abstracts_

### Official Review · Reviewer_DXzD · 2025-10-24

**Soundness:** 4
**Correctness:** 4
**Rating:** 5
**Confidence:** 3

**Summary:**

The authors aim to develop a Maori LLM, and present the necessary steps and challenges. They motivate this endeavour by the large performance gaps of LLMs for languages where a lot of data is available, and languages with less data. This could leave speakers of smaller languages behind.
The authors acknowledge risks like data poisoning, and highlight that they will include the Maori community.

**Strengths:**

- The abstract is written in clear language and is easily understandable for non experts.
- The authors motivated their project thoroughly and it is evident why their work is significant.
- The steps and challenges were presented in a coherent way. Overall the challenges, goals and methods are as clear as they can be presented on two pages.
- The content seems correct to me.

**Weaknesses:**

- Efforts on other small languages are mentioned, is there something we can learn from them? Maybe try to connect and compare them to your work.
- Could you quantify the available resources and how they compary to those of other languages? Especially the balance between audio and written language.

---

### Official Review · Reviewer_KJic · 2025-10-27

**Soundness:** 2
**Correctness:** 3
**Rating:** 2
**Confidence:** 3

**Summary:**

In this abstract, the author presents an early-stage plan for a Large Language Model for Te Reo Maori. In particular, the author aims to curate and clean the training data (with the possibility of generating synthetic text with a knowledge graph), optimize the tokenization process, and implement and evaluate a model. All this will be done with the participation of CARE and Maori language speakers and iwi.

**Strengths:**

The work aims to address an issue that is relevant to the preservation of indigenous languages. Furthermore, as stated by the authors, this work would be the first specific study on the Te Reo Maori language. To support the development plan, the author has identified 500 million Maori words, the problem of tokenization, and the author plans to involve the Maori language community to collect training examples, build benchmarks, and perform model evaluation. As a possible development, this research could help guide other similar cases involving low-resource languages.

**Weaknesses:**

Although the idea seems promising, I believe it is still somewhat vague and in its early stages. In particular, as the author points out, the following are unclear:
- The training dataset, with many other datasets still to be checked (lines 101-104);
- The model that will be used and the fine-tuning process;
- The tokenisation process;
- How the quality and authenticity of indigenous texts will be evaluated (lines 83-84).

Section 3.2 is the only one that provides details or ideas on the strategies to be adopted. Consequently, in the absence of a minimum of details, I consider the abstract premature. I am open to reconsidering my position if a non-technical abstract is acceptable to the conference, and I leave the opinion to the other reviewers and AC.

---

### Official Review · Reviewer_pKy3 · 2025-11-04

**Soundness:** 3
**Correctness:** 3
**Rating:** 4
**Confidence:** 4

**Summary:**

This abstract outlines a plan to develop a sovereign Large Language Model (LLM) for Te Reo Māori, aiming to bridge the performance gap between high- and low-resource languages. The proposed work emphasizes community-led data collection, culturally aligned benchmarks, tokenizer adaptation for Māori orthography, and adherence to CARE principles for Indigenous data governance. The goal is to create foundational datasets, evaluation tools (MāoriGLUE), and best practices that enable ethical and effective Māori NLP development.

**Strengths:**

Timely and Important Topic:
The abstract addresses the growing inequity in LLM performance across languages, focusing on linguistic inclusion and digital sovereignty — an area of increasing global relevance.

Strong Ethical Framing:
The commitment to CARE principles and Māori community governance demonstrates a mature understanding of the ethical dimensions of AI and Indigenous data use. This grounding gives the project legitimacy and long-term social value.

Clear and Coherent Vision:
Despite its brevity, the abstract effectively articulates a structured roadmap — from corpus creation to model evaluation — while maintaining readability and conceptual flow.

Potential for Broader Impact:
If realized, this project could serve as a model for other Indigenous and low-resource language communities, establishing reproducible methodologies for culturally responsible LLM development.

Appropriate for Abstract Track:
The work is exploratory and proposal-oriented rather than conclusive, making it well-suited to a discussion-focused venue such as an abstracts or works-in-progress session.

**Weaknesses:**

Limited Technical Specificity:
As expected in an abstract, methodological detail is light. However, brief mention of intended model scales, expected datasets, or partnership structures could improve clarity.

Feasibility Unclear:
The abstract could better indicate resource scope (compute, personnel, funding) or specific collaborations to ground the plan in practical terms.

Evaluation Plan Remains Vague:
While MāoriGLUE is a compelling idea, the types of tasks or metrics envisioned are not described; one or two examples would strengthen the proposal’s concreteness.

These are not critical flaws for an abstract but areas that could benefit from clarification during presentation or discussion.

---

### Decision · Program_Chairs · 2025-11-05

**Decision:**

Accept

**Comment:**

The reviewers found the abstract borderline, yet the PCs believe it will be of interest to the community and should have the opportunity be presented.